# Dysregulation of ErbB4 Signaling Pathway in the Dorsal Hippocampus after Neonatal Hypoxia-Ischemia and Late Deficits in PV^+^ Interneurons, Synaptic Plasticity and Working Memory

**DOI:** 10.3390/ijms24010508

**Published:** 2022-12-28

**Authors:** Harisa Spahic, Pritika Parmar, Sarah Miller, Paul Casey Emerson, Charles Lechner, Mark St. Pierre, Neetika Rastogi, Michael Nugent, Sarah Ann Duck, Alfredo Kirkwood, Raul Chavez-Valdez

**Affiliations:** 1Division of Neonatology, Department of Pediatrics, Johns Hopkins University School of Medicine, Baltimore, MD 21287, USA; 2Mind-Brain Institute, Department of Neuroscience, Johns Hopkins University School of Medicine, Baltimore, MD 21205, USA; 3Department of Molecular and Cellular Biology, Johns Hopkins University, Baltimore, MD 21205, USA

**Keywords:** parvalbumin, neuregulin-1, long-term depression, long-term potentiation, Akt

## Abstract

Neonatal hypoxic-ischemic (HI) injury leads to deficits in hippocampal parvalbumin (PV)^+^ interneurons (INs) and working memory. Therapeutic hypothermia (TH) does not prevent these deficits. ErbB4 supports maturation and maintenance of PV^+^ IN. Thus, we hypothesized that neonatal HI leads to persistent deficits in PV^+^ INs, working memory and synaptic plasticity associated with ErbB4 dysregulation despite TH. P10 HI-injured mice were randomized to normothermia (NT, 36 °C) or TH (31 °C) for 4 h and compared to sham. Hippocampi were studied for α-fodrin, glial fibrillary acidic protein (GFAP), and neuroregulin (Nrg) 1 levels; erb-b2 receptor tyrosine kinase 4 (ErbB4)/ Ak strain transforming (Akt) activation; and PV, synaptotagmin (Syt) 2, vesicular-glutamate transporter (VGlut) 2, Nrg1, and ErbB4 expression in coronal sections. Extracellular field potentials and behavioral testing were performed. At P40, deficits in PV^+^ INs correlated with impaired memory and coincided with blunted long-term depression (LTD), heightened long-term potentiation (LTP) and increased Vglut2/Syt2 ratio, supporting excitatory-inhibitory (E/I) imbalance. Hippocampal Nrg1 levels were increased in the hippocampus 24 h after neonatal HI, delaying the decline documented in shams. Paradoxically ErbB4 activation decreased 24 h and again 30 days after HI. Neonatal HI leads to persistent deficits in hippocampal PV^+^ INs, memory, and synaptic plasticity. While acute decreased ErbB4 activation supports impaired maturation and survival after HI, late deficit reemergence may impair PV^+^ INs maintenance after HI.

## 1. Introduction

Fast-spiking parvalbumin-expressing (PV^+^) interneurons (INs) are inhibitory GABAergic cells essential to preserving the excitatory/inhibitory (E/I) balance [1,2], and mechanisms of synaptic plasticity, particularly long-term depression (LTD) in the hippocampus [3]. In the hippocampus, decreased inhibitory inputs from PV^+^ INs to pyramidal cells leads to neuronal hyperexcitability, impaired memory, inability to focus, and decreased impulse control [4,5]. We have reported that hypoxia-ischemia (HI) in the neonatal mouse (P10) results in ~40% decrease in the number and dendritic complexity of PV^+^ INs in the dorsal hippocampus 8 days after the insult [6,7,8]. Neonatal HI leads to memory and cognitive deficits which persist despite therapeutic hypothermia (TH), the only clinically available treatment for infants suffering of HI injury and encephalopathy (HIE) [9,10,11]. The incomplete protection provided by TH to memory domains of neurobehavior in the mouse and the limited protection to the hippocampus and mirrors the reports from randomized clinical trials in humans [12]. Thus, persistent deficit of PV^+^ INs after neonatal HI may explain worse neurodevelopmental outcomes in humans, but the mechanisms leading to this deficit, and thus potential therapeutic targets, are still understudied.

Because the number of hippocampal PV^+^ INs becomes deficient only 8 days after the HI insult secondary to the lack of increase in their numbers, we have speculated that mechanisms controlling the maturation of PV^+^ INs were involved. Hippocampal GABAergic INs mature postnatally in mice and in humans [13]. Neuregulin (Nrg)-1 supports migration, maturation, and survival of PV^+^ INs [14,15,16,17,18]. Nrg1 activates the erb-B2 receptor tyrosine kinase 4 (ErbB4) to guide tangential migration, dendritic development, axonal elongation, and synaptic plasticity [19,20,21,22,23,24,25,26,27]. ErbB4 acts on phosphoinositide (PI) 3 kinase/ Ak strain transforming (Akt) which causes downstream effects to support PV^+^ IN dendritic arborization [28,29]. Genetic deletion of *ErbB4* in the hippocampus reduces PV^+^ IN counts [30], supporting the importance of ErbB4 in the maintenance of the PV^+^ IN population [31,32]. The role of Nrg1/ErbB4 in the mechanisms of injury and repair have been studied in other brain injury paradigms [33,34]. Here, we hypothesized that neonatal HI leads to persistent deficits in PV^+^ INs, working memory and synaptic plasticity associated with ErbB4 dysregulation and that TH does not attenuate significantly these effects.

## 2. Results

### 2.1. Deficit in PV^+^ IN Counts in Dorsal Hippocampus after Moderate-Severe Neonatal HI Persists to Early Adulhood

We have previously reported that the modified Vannucci model with 45 min of hypoxia exposure, produces an overall mortality of ~20%, severe injury with hippocampal obliteration in ~20% of the operated mice and minimal injury in ~12–15% of those mice [6,9,10]. Excluding animals with obliterated hippocampus or those with no injury, we have documented that moderate-severe HI injury produces significant deficits in the number of PV^+^ INs in the dorsal hippocampus, which emerge at P18 (~2 year old human equivalent) and is not attenuated by TH [6]. Current experimental design is shown in Figure 1A. Now, we show that at P40 (adolescent-early adulthood equivalent in humans), HI-injured mice had ~50% lower number of hippocampal PV^+^ IN than sham mice (Figure 1B_1_, C). Again, these deficits were similar in HI-injured mice regardless of sex or TH (Figure 1B_2_). These HI-injured mice also had smaller residual hippocampal volumes, which correlated with GFAP-derived injury severity score (*p* < 0.001; Figure 1D_1_) [6,7,35]. However, PV^+^ IN counts did not correlate with injury severity at P40 (Figure 1D_2_), similar to what was reported at P18 [6]. 

### 2.2. Spatial Working Memory Impairments after Neonatal HI and Correlation with Deficits in PV^+^ IN Counts in Dorsal Hippocampus

HI-injured mice: (i) spent less time in the center of the open field (OF) arena (*p* = 0.04 NT vs. sham and *p* = 0.03 TH vs. sham [KW ANOVA *H* (2) = 7.92, *p* = 0.01], Figure 2A_1_,A_2_), (ii) spent less time exploring the novel arm during Y-maze (YM) phase 2 (*p* = 0.04 NT vs. sham and *p* = 0.001 TH vs. sham [KW ANOVA *H* (2) = 13.48, *p* = 0.001], Figure 2B_1_,B_2_), and (iii) have lower object location task (OLT) exploratory preference (EP, *p* = 0.008 NT vs. sham and *p* = 0.04 TH vs. sham [KW ANOVA *H* (2) = 9.72, *p* = 0.008], Figure 2C_1_,C_2_) compared to shams, supporting anxiety-like behavior and deficits in working memory and spatial learning not fully addressed by TH. Stratification by sex demonstrated more robust effects of HI in female mice spending less time in the YM novel arm (KW ANOVA *H* (2) = 10.95, *p* = 0.004, *n* = 10–12/group) and having lower OLT EP (KW ANOVA *H* (2) = 6.22, *p* = 0.04, *n* = 10–12/group), while HI-injured male mice demonstrated not significant trends (*n* = 10–14/group, not shown). Deficits in PV^+^ IN counts in the dorsal hippocampus weakly but significantly correlated with lower (i) time in OF center (rho = 0.37, *p* =0.008, Figure 2A_3_), (ii) time exploring the YM novel arm (rho = 0.37, *p* = 0.01, Figure 2B_3_) and (iii) OLT EP score (rho 0.47, *p* < 0.001, Figure 2C_3_). Thus, deficits in PV^+^ INs may in fact be a significant contributor to the behavioral deficits seen in the model and not addressed by TH. 

### 2.3. Disturbed Mechanisms of Synaptic Plasticity 8 and 30 Days after Neonatal HI

#### 2.3.1. Persistently Blunted LTD in Schaffer Collateral Synapse after Neonatal HI

Because PV^+^ INs deficits will likely result in decreased local inhibition to pyramidal cells, then, we hypothesized that long-term depression (LTD) would be disrupted in extracellular field potentials (EcFP) measured in the Schaffer collateral synapse (CA3→CA1) of HI-injured hippocampi compared to controls. Since TH did not provide significant protection against deficits in PV^+^ INs count and behavioral performance, as showed here, electrophysiology experiments were focused in the understanding of the effects of HI injury, using sham hippocampus (separate set of animals) and hypoxia-alone exposed hippocampus (left hemisphere, contralateral to the HI injury) as controls for comparisons. Sham and hypoxia-exposed hippocampi showed preserved LTD, with 40.8% (IQR, −56.5% to −26.5%) and 39.0% (IQR, −59.6% to −28.2%) decrease in field potential (FP) slope from baseline, respectively at ~P18 (P17-P19) (Figure 3B,C_1_, *p* < 0.05 by Wilcoxon test vs. baseline), and 58.5% (IQR, −61.0% to −50.3%) and 42.7% (IQR, −49.3% to −31.3%) decrease, respectively at ~P40 (P38-P42) (Figure 3B,C_2_, *p* < 0.05 by Wilcoxon test vs. baseline). In contrast, HI-injured hippocampi LTD was blunted at ~P18 (KW *H* (2) 15.65, *p* < 0.001; *p* = 0.005 vs. sham & *p* = 0.002 vs. hypoxia, *n* = 6–10/group) and at ~P40 (KW *H* (2) 19.16, *p* < 0.001; *p* < 0.001 vs. sham & *p* = 0.006 vs. hypoxia; *n* = 6–10/group). No sexual dimorphism was identified.

#### 2.3.2. Preserved Schaffer Collateral LTP in Male Hippocampus Contrasts with Heightened LTP in Female Hippocampus 30 Days after HI

Although LTD was blunted 30 d after neonatal HI, long-term potentiation (LTP) was preserved in both sexes, although dimorphism existed in magnitude of response to the stimuli. Schaffer collateral stimulus-evoked LTP was similar in sham male and female mice (% change of 94% and 80.2% from baseline, respectively; Figure 3D_1_, white boxes). Hypoxia-exposed hippocampi trended to have attenuated stimulus-evoked LTP (% change of 49.3% in males and 51.1% in females, Figure 3D_1_, light grey boxes). A similar trend for attenuated LTP was documented in male HI-injured hippocampi, (63%, IQR 34.2–86.8%, dark grey box). In contrast, the magnitude of FP change in female HI-injured Schaffer collateral synapse was heightened with a change of 216.3% from baseline (Figure 3D_1_, dark grey boxes; KW *H* (2) 12.9 *p* = 0.002, *p* = 0.04 vs. sham, *p* < 0.001 vs. hypoxia). 

#### 2.3.3. Increase in Vglut2/Syt2 Ratio in HI-Injured Dorsal Hippocampus Supports Relative Deficit in PV Inhibitory Input

PV^+^ INs deliver local inhibitory inputs to soma and proximal dendrites of pyramidal cells modulating their synaptic activity. Puncta expressing Syt2, a pre-synaptic GABAergic marker specific to PV^+^ INs and puncta expressing Vglut2, a pre-synaptic glutamatergic marker, were quantified using Imaris software (Oxford instruments) to use as a surrogate of excitatory/inhibitory (E/I) balance within the CA3 subfield (Schaffer collaterals). Syt2 was decreased by 16.6% and 20.1% in male and female CA3 pyramidal cell layer (Py) (*p* < 0.05 vs. sham), respectively (Figure 4A). Sexual dimorphism in synaptic compensation existed as Syt2^+^ puncta volume was 39.3% larger in males, but not in female HI-injured CA3. (Figure 4B, MWU *p* = 0.03 vs. sham), which resulted in decreased Syt2^+^ expression (immunofluorescent intensity, IF) within CA3 Py puncta (Figure 4C) only in female mice (*p* = 0.01 vs. sham). The ratio between the excitatory Vglut2 and the inhibitory Syt2 thus suggested an excitatory predominance only in the CA3 Py of HI-injured female mice (Figure 4D,E; *p* = 0.03), which matches the heighted LTP and the blunted LTD also documented in the female HI-injured Schaffer Collateral synapse.

### 2.4. Persistent Dysregulation of Nrg1/ErbB4 Pathway after Neonatal HI Injury of the Hippocampus

ErbB4 and its ligand Nrg1 have been linked to mechanism leading to migration, maturation and survival of PV^+^ INs [28], thus mechanistically we hypothesized that dysregulation in the Nrg1/ErbB4 pathway may explain the persistent deficit in the number of hippocampal PV^+^ INs in response to neonatal HI injury. 

#### 2.4.1. Delayed Developmental Decline in Hippocampal Nrg1 Levels in Response to Neonatal HI

PV expression in medial ganglionic eminence-derived INs occurs late in their maturation [36]. Nrg1 promotes excitatory synapse development by stabilizing the assembly of PSD95-ErbB4 on PV ^+^ INs, modulating GABAergic activity as well as PV^+^ IN maturation [16,19,25,26,27,28,37,38,39,40,41]. Here, we document that developmentally Nrg1 levels decreased by ~50% in the mouse hippocampus after P11 reaching a nadir at or after P15 (R^2 (quadratic)^ 0.72, *p* < 0.001; Figure 5A), temporarily coinciding with the peak of hippocampal PV^+^ IN maturation. Neonatal HI increased Nrg1 levels by 84.4% (*p* = 0.01, NT vs. sham; KW *H* (2) 9.72, *p* = 0.008) and 44.8% (*p* = 0.05, NT vs. sham; KW *H* (2) 5.92, *p* = 0.04) at 24 h (P11) and 120 h (P15) post-insult, respectively (Figure 5B), with levels being similar to those in sham thereafter irrespective to treatments (Figure 5B). αII-fodrin breakdown products (BDP) were measured to correlate severity of injury with acute Nrg1 rise after HI at P11. αII-fodrin 150 kDa BDP (necrotic-like cascades) did correlate with Nrg1 (Spearman’s Rho r = 0.53, *p* = 0.004), relationship was best represented by a cubic regression line (*y* = 13.18 + 41.06**x* − 33.74**x*^2^ + 8.25**x*^3^), which plateaued at Nrg1 level of ~28.5 pg/gr (Figure 5C_1_) with representative Western blots shown in Figure 5C_2_. Unlike males (Figure 5D_1_), HI-injured female mice treated with TH only sustained elevated hippocampal Nrg1 levels for 24 h after HI (KW *H* 7.43, *p* = 0.02; Figure 5D_2_). Abundant Nrg1 expression (red channel) within the Py of the dorsal CA1 (Figure 5E_1_) and CA3 (Figure 5E_2_) contrast with low PV expression (green channel). Thus, acute increase in hippocampal Nrg1 level after neonatal HI may be an attempt to compensate for excitotoxicity.

#### 2.4.2. Acute and Late Deficits in ErbB4 Receptor Expression and Activation after Neonatal HI

The increase in Nrg1 levels at 24 h after HI contrasted with the decreased overall ErbB4 phosphorylation (KW *H* (2) 6.41, *p* = 0.04, Figure 6A), which resulted from the decreased total ErbB4 levels (−36%, *p* = 0.003 NT vs. Sham; KW *H* (2) 14.48, *p* = 0.04 = 01; Figure 6B) and leaded to preserved ratios (Figure 6C). The decrease in ErbB4 expression 24 h after HI was associated with increased 150 kDa αII-fodrin BDP (R^2^ (cubic) 0.78, *p* < 0.001; y = 1.06 − 0.89*x +0.33*x^2^ − 0.04*x^3^; Figure 6D), and increased Nrg1 (R^2^ 0.42, *p* = 0.001; y = 38.96 − 21.96*x; Figure 7E), supporting acute Nrg1 changes as a possible compensation for the lost in ErbB4. By P15 and P18, ErbB4 phosphorylation and total expression returned to levels similar to those in sham hippocampus (Figure 6A,B). This ‘recovery’ in ErbB4 expression coincided with increased expression on surviving PV^+^ INs demonstrating evidence of stagnant maturation with low PV expression and simplified dendritic arbors or ongoing cell death (Figure 6F). Deficits remerged at P40, with a 43% (*p* <0.001 NT vs. sham; KW *H* (2) 17.85, *p* < 0.001) and 22% (*p* = 0.02 NT vs. sham; KW *H* (2) 16.97, *p* < 0.001) lower ErbB4 phosphorylation (Figure 6A) and total expression (Figure 6B), respectively, resulting in a 24% lower ErbB4 phosphorylated to total ratio (KW *H* (2) 13.94, *p* =0.001; Figure 6C). While TH prevented the decrease in total ErbB4 levels in P40 HI-injured hippocampus (*p* < 0.001 TH vs. NT mice, Figure 6B), this was not enough to prevent the decrease in phosphorylation (Figure 6A,C). These responses were similar in both sexes. Phosphorylated and ratio to total ErbB4 correlated with time spent in the center of the open field arena as well as the total travel time in OLT (Appendix A).

#### 2.4.3. Akt-Dependent Cascade after Neonatal HI

Akt is a non-cell type-specific adaptor involve in pro-survival cascades via inhibition of apoptosis. Nrg1 via binding to ErbB4 receptor leads to recruitment of Akt leading to survival [42], and dendritic development of PV^+^ INs [43]. Thus, we hypothesized that deficits in ErbB4 expression and activation would result in decreased Akt phosphorylation, which temporarily will precede deficits of PV^+^ INs, and later lead to dendritic simplification as previously reported by us [6,8]. Indeed, we documented that Akt phosphorylation was decreased by 39.1% in NT mice (*p*= 0.02) and 61.5% in TH mice (*p* = 0.004) compared to sham mice at P11 (KW *H* (2) 9.41, *p* =0.009; Figure 7A_1_). Deficits in pAkt and pAkt/Akt ratio at P11 (Figure 7A_1_,A_2_) were directly correlated with ErbB4 phosphorylation (*y* = 0.12 + 1.27**x* [R^2^
_(linear)_ 0.59] and *y* = 0.27 + 0.94**x* [R^2^
_(linear)_ 0.66], respectively; *p* < 0.001; Figure 7B_1_,B_2_). No sexual dimorphism existed in these relationships at P11. Unlike the relationships described at P11, we found no difference in pAkt between NT and sham (Figure 7C) contrasting with an increased pAkt (47.6%, *p* = 0.02 vs. sham; 106%, *p* = 0.001 vs. NT) in the P40 HI-injured hippocampus treated with TH (KW *H* (2) 11.09 *p* = 0.004; Figure 7C). ErbB4 phosphorylation predicts significantly (*p* = 0.004) but weakly the changes in pAkt at P40 (R^2^
_(linear)_ 0.16; Figure 7D). Sex stratification additionally exposed that the increase in pAkt in TH mice occurred more robustly in females at P40 (Figure 7E). 

## 3. Discussion

Here, we show for the first time the nature of ErbB4 deficits in the HI-injured hippocampus to provide a mechanistic context to the persistent deficit of PV^+^ INs, disruption of mechanisms of synaptic plasticity and behavioral correlates. We have previously reported that hippocampal PV^+^ INs are decreased as early as 8 d after neonatal HI and that TH does not prevent this deficit [6,7,8]. Here, we show that this deficit may persist at least to 30 d after neonatal HI (human equivalent to adolescence), and correlates with spatial working memory deficits and anxiety-like behavior. Persistently impaired stimulus-evoked LTD in the Schaffer Collateral synapse after neonatal HI, provides a electrophysiology correlate to memory deficits, while the heightened stimulus-evoked LTP and excitatory predominance in female mice may explain previous reports of anxiety-like behaviors after neonatal HI particularly in female mice [9,10,11], which have been also associated with history of neonatal HI encephalopathy in humans [44]. Mechanistically, it is not surprising that acute Nrg1 increase in the HI-injured hippocampus does not result in increased ErbB4 activation, due to the decrease in receptor levels linked to the degree of hippocampal necrosis. The correlation between acute decreased activation of ErbB4 and Akt 24 h after neonatal HI supports impaired mechanisms of survival. Thus, in this context, the acute increase in Nrg1 is likely a compensation for the loss of ErbB4 early after HI. ErbB4 overexpression on PV^+^ INs surviving 8 d after neonatal HI may be a mechanism of compensation explaining the recovery in receptor expression after the injury. However, ErbB4 deficits reemerging 30 d after neonatal HI despite treatment with TH may explain delayed neurodegeneration, as we have reported previously [7]. Since deficits in working memory and anxiety-like behaviors have been previously linked to deficits in the ErbB4 in PV^+^ INs [37], thus, we postulate that the acute and late ErbB4 deficits in the hippocampus after neonatal HI may be the mechanism leading to persistent deficits in PV^+^ INs in the dorsal hippocampus after and contributing in part to the impaired memory and mechanisms of synaptic plasticity after HI. 

Decreased PV^+^ IN counts in the dorsal hippocampus 30 d after neonatal HI demonstrates the persistence of the deficits first identified 8 d after the insult [6]. TH likely does not prevent either PV^+^ IN or behavioral deficits in this model in agreement with previous reports [9,10,11]. Although most RCTs for TH were underpowered to evaluate cognition and memory, there is enough evidence to support that cognitive, memory, and behavioral deficits are also not fully addressed by TH either in humans [12,45,46,47]. To understand some of the mechanisms behind these persistent deficits, we report for the first time a correlation of anxiety-like and working memory deficits with decreased PV^+^ IN count in the dorsal hippocampus. However, it important to emphasize that because preserved, but injured dorsal hippocampi were the target for evaluation in our experiments, hippocampi with GFAP injury scores <9 or complete obliteration were not included in these analyses thus decreasing the variability of behavioral outcomes and strengthening correlations. Furthermore, hippocampi and pups demonstrating none to mild injury were removed from analysis to correspond to clinical practice where moderate to severe HI injury/encephalopathy is the standard criteria for initiation of TH in newborns suffering of HI injury of the brain. The proposed link between functional disruption of PV^+^ INs in the hippocampus and memory impairment is corroborated by other studies demonstrating that loss of inputs to CA1 PV^+^ INs or their functional inhibition alteration results in memory deficits [48,49,50]. GABAergic deficits, specifically loss of hippocampal PV^+^ INs, have been associated with models of neuropsychiatric [51,52,53] and neurodegenerative disorders [54,55,56]) in rodents. Similar associations have been described in postmortem human studies [57,58,59,60,61]. Thus, the long-term effects of neonatal HI in models of schizophrenia and Alzheimer’s disease merit investigation. 

Hippocampal PV^+^ INs surviving HI develop somatodendritic attrition, with simplified dendritic arbors [6,7,8]. GABAergic axonal terminals (GAD65/67^+^) perisomatic to pyramidal cells are also decreased in the CA1 and CA3 after neonatal HI [6,7]. As shown here, Syt2^+^ axonal terminals, pre-synaptic exclusive from PV^+^ INs, represent a portion of those decreased GABAergic axonal terminals reduced by HI equally in both sexes. However, in the HI-injured CA3 of male mice, the volume of Syt2^+^ axonal terminals are larger, perhaps preventing the E/I imbalance towards excitation documented in HI-injured females CA3. Not surprisingly, the molecular mechanisms of synaptic plasticity measured using EcFP in the Schaffer collateral synapse are also disrupted, with blunted stimulus-evoked LTD in both sexes, and heightened LTP magnitude only in female mice 30 d after HI. To our knowledge, this is the first time that mechanisms of synaptic plasticity have been studied in a model of neonatal HI, due to the technical difficulties. Previous studies in models of chronic hyperoxia from P2–14 have also demonstrated sexually dimorphic heightened LTP magnitude in the Schaffer collateral synapse [62]. Conversely, intermittent hypoxia from P3–7 in C57BL6 mice results in decreased LTP at P42 [63], which may provide support to the trend documented in hypoxia-exposed hippocampus (contralateral to HI) in our experiments. 

Mechanistically, we have focused on the study of the Nrg1-ErbB4 signaling pathway as it is required for PV expression [64] and dendritic arborization [28,29] in the maturation of medial ganglionic eminence-derived INs. Changes in Nrg1 and ErbB4 expression mainly relate to PV^+^ INs, [31,32] as such ErbB4 genetic deletion causes decreased PV^+^ IN counts [30]. Similarly, decreased hippocampal ErbB4 either throughout development or during adulthood disrupts activity in OF and contextual [65] and working memory [37], worse if the onset of ErbB4 deficits is during early post-natal hippocampal development. Altogether, the ErbB4 signaling pathway rises as a plausible mechanism by which neonatal HI results in deficits in hippocampal PV+ INs and memory impairments. We report that Nrg1 acutely increases in the hippocampus following HI injury, but ErbB4 activation is decreased due to low ErbB4 protein expression, which is not addressed by TH. Although Nrg1 may be neuroprotective [21] by attenuating apoptosis [66]; these effects are only afforded if associated with ErbB4 [33,67]. Akt has also been targeted to prevent cell death [68], but the lack of ErbB4 activation and Akt phosphorylation acutely (24 h) after neonatal HI suggest a failed attempt to prevent cell death. Interestingly, deficits in ErbB4 expression and activation are variable, with a second period of deficit detected as late as 30 d after HI (P40) a time at which, we have reported significant neurodegeneration of PV^+^ INs [6,7], thus suggesting that delayed deficit in ErbB4 activation may impair the maintenance of these cells in the HI-injured hippocampus. 

### Limitations

Some inherent limitations exist. First, inability to directly correlate broad biochemicals (i.e., Nrg1, ErbB4 in the whole hippocampus) to specific IHC findings (i.e., PV^+^ INs in the hippocampal subfields) limit direct associations. Second, because preserved but injured dorsal hippocampus was the target for evaluation in our experiments for correlation with PV^+^ INs, thus lack of hippocampal injury by GFAP scoring (scores < 9) or complete obliteration by HI were not included in these analyses, decreasing the common variability of the behavioral outcomes. Thus, our behavioral results do not demonstrate the full spectrum of variability of neurobehavioral outcomes that are observed with the Rice-Vannucci model of neonatal HI as are better represented elsewhere [9,10]. Experiments to salvage early and late ErbB4 deficits after neonatal HI are warranted to determine whether these relationships are corroborated

## 4. Materials and Methods

### 4.1. Animals and Experimental Design

ARRIVE guidelines were followed as approved by the Institutional Animal Care and Use Committee at Johns Hopkins, following the NIH, US Department of Health and Human Services 85–23, 1985. The C57BL6 mice (Charles River Laboratory; Newark, NJ, USA, DE, litter size = 5–6 pups) were randomized 2:1 to HI and sham groups (Figure 1A). Pups received isoflurane for anesthesia (3% induction until incision and 1% maintenance) and anesthesia-exposed littermates served as controls. Cerebral HI (Rice-Vannucci model modified for mice) was produced at postnatal day (P) 10 followed by hypoxia exposure (FiO_2_ = 0.08 at 36 °C for 45 min) [6]. HI-injured mice were randomized 1:1 to TH (31 °C) or normothermia (NT, 36 °C) for 4 h. Core body temperatures were monitored with rectal thermocouple microprobe (Ad Instruments, Inc., Colorado Springs, CO, USA). This preclinical model of TH provides neuroprotection sustained in the cortex, but transient in the hippocampus [9,10]. A total of 292 mice (51 litters) were used for the experiments (31 mice for P11, 32 mice for P15, 59 mice for P18, and 170 mice for P40). Figures show the number of animals included in the analysis at each time point. The mortality rate was 20.1% after HI procedure, thus 179 mice survived after HI equally distributed by sex. 

### 4.2. Brain Processing

Mice were sacrificed P11, P15, P18, and P40 using inhaled isoflurane. Mice assigned to immunohistochemistry (IHC) received a 10 mL trans-cardiac infusion of 0.1 M PBS pH 7.4 followed by perfusion with 4% paraformaldehyde (PFA)/0.1 M PBS for 5 min (4 mL/min) and post-fixation overnight in 4% PFA for mice ≤ P18 and for 2 h for P40. Brains were cryoprotected in sucrose gradient until sinking, snap-frozen in dry-ice cold 2-methylbutane, and stored at −80° C for later coronal sectioning (50 µm) using freezing microtome to study the dorsal hippocampus [6,7]. For electrophysiology experiments mice underwent isoflurane anesthesia, decapitation, brain isolation and vibratome sectioning (acute 300 µm-thick slices with dorsal hippocampus) for extracellular field potentials (EcFP). For fresh tissue collection mice were decapitated after anesthesia, and brains were transferred onto a freezing plate where the hippocampus proper was micro-dissected, and snap-frozen and stored at −80 °C.

### 4.3. Immunohistochemistry (IHC)

Floating IHC was performed for GFAP and PV DAB-based IHC and multichannel immunofluorescence (IF) experiments as reported previously [6,7,8]. 

#### 4.3.1. GFAP-Derived Injury Scoring System

A semi-quantitative GFAP-derived scoring system was used to assess hippocampal injury [6,7,8,11,35], using the rabbit IgG anti-GFAP antibody (DAKO North America, Inc., Carpinteria CA, USA, Z0334; RRID:AB_10013382) at 0.4 µg/mL. Morphological changes in astrocytic size, number and thickness of branching, presence of glial scaring and overlap between astrocytic domains were evaluated [6,8,35]. The sum of subfield (CA1, CA3, and DG) scores (range:1 to 8) in the dorsal and ventral hippocampus produced an accumulative score maximum Score: 48). 

#### 4.3.2. DAB-Based IHC for PV

Sections were washed 10 min × 3 times in TBS, incubated in methanol with H_2_O_2_ (45 min at 4 °C), permeabilized with 0.6% triton X/TBS, and blocked with 10% normal goat serum (NGS)/0.1% Tween/TBS (TBS-T). Tissues were incubated overnight with primary rabbit IgG anti-PV antibody (Novus Biologicals, NB120-11427; RRID:AB_791498) at 1:250 (4 µg/mL), followed by goat anti-rabbit antibody (1:200) in 4% NGS and DAB stain [6]. PV counts were not determined in the 300 μm acute slices used for electrophysiology as they were not suitable for IHC. 

#### 4.3.3. Multichannel Immunofluorescent (IF)-IHC

Coronal sections containing dorsal hippocampi were selected, washed in TBS pH 7.2 (10 min × 3), incubated in sodium citrate buffer pH 6.0 for antigen retrieval (80 °C × 90 min), permeabilized in triton X in TBS (×15 min; 0.2% for P11 and P15, 0.4% for P18, or 0.6% for P40), and blocked in 10% NGS/TBS-T (60 min). Primary antibody exposure overnight (4 °C), was followed by secondary antibody mix in 4% NGS/TBS-T for 2 h (Appendix A). Secondary antibodies are conjugated Alexa Fluor 488 (green), 568 (red) and 647 (deep red), with nuclei stained using 4,6-diamidino-2-phenylindole (DAPI, 1 μg/mL) in TBS for 5 min prior to TBS washing and ProLong Glass Antifade (Thermo Fisher Scientific, Waltham, MA, USA; P36980) mounting.

#### 4.3.4. Primary Antibodies for IF-IHC (Appendix A)

(i)PV (Novus, Littleton, CO, USA, NBP2-50036; RRID: AB_2814697) Chicken polyclonal IgY antibody raised against recombinant full-length human PV (2.5 μg/mL)(ii)Nrg1 (ProteinTech Group, Rosemont, IL, USA, 66492-1-Ig; RRID: AB_2881857) Mouse monoclonal IgG_1_ antibody raised against NRG1 Fusion Protein [ProteinTech Group Arg0803] (4 μg/mL)(iii)ErbB4 (GeneTex, Irvine, CA, USA, GTX80811; RRID: AB_625545) Mouse monoclonal IgG_2b_ antibody raised against an unconjugated synthetic peptide encompassing amino acids 1250-1264 (3 μg/mL)(iv)Syt2 (Developmental Studies Hydridoma Bank, Iowa City, IA, USA, 808402; RRID: AB_2315626) Mouse monoclonal IgG_2a_ antibody raised against zebrafish synaptotagmin-2 [69](v)VGlut2 (Abcam, Cambridge, UK, ab216463; RRID: AB_2893024) Rabbit monoclonal IgG antibody raised against synthetic VGlut2 peptide

#### 4.3.5. Negative Controls

The specificity of all antibodies used in IF-IHC experiments were determined by immunoblotting using NGS blocking (Appendix A). All experiments were run with negative controls. Negative controls were either not exposed to primary antibody or exposed to the species/subtype specific immunoglobulin at similar concentrations to those used for primary antibodies.

### 4.4. Confocal Microscopy and Image Processing

Z-stacks were taken at 1440 × 2880 pixels, 16-bit depth, and averaged ×2, captured at 63×/1.40 oil DIC objective and 1.0 zoom to produce uncompressed images of 101.61 × 203.22 μm presenting the CA3 subfields (2 stitched high magnification fields). Z-stacks were set at 1 air unit (0.9 µm) to the maximal wavelength (639 nm) in a Laser Scanning Confocal Microscope LSM700 AxioObserver (Carl Zeiss AG, Oberkochen, Germany). Same pinhole, gain and offset settings were used for all repeats. Z-stacks were saved in .czi, and converted to .ims for image processing in Imaris ×64 v9.7.2 software (Bitplane, Belfast, UK). An algorithm was created to determine channel source, level of surface detail, creation threshold, and sphere size. DAPI-labeled nuclei were used to delineate the pyramidal cell (Py) layer. Puncta count, volumes, and IF intensity data were analyzed.

### 4.5. Semi-Quantitative Assessment of Hippocampal Atrophy

As detailed previously [7,8,11], hippocampal areas were measured in Nissl-stained 50 μm sequential coronal sections 600 µm apart (antero-posterior axis). Hippocampal volumes were extrapolated using the formula: $$\mathrm{Hippocampal}\;\mathrm{volume}\;\left({\mathrm{mm}}^{3}\right)=\sum_{i=1\;\left(n=1\right)}\left[S_{i}*0.05\right]+\sum_{i=1\;\left(n=1\right)}[\left(S_{i}+S_{i+1}\right)*0.3]\;$$

S = hippocampal area (mm^2^)I = section position in anterior-posterior axisn = number of sum repeats

Residual ipsilateral hippocampal volume (%) was calculated relative to contralateral hippocampus. 

### 4.6. Nrg1 Enzyme-Linked Immunosorbent Assay (ELISA)

Hippocampal tissues underwent extraction protocol using an acid-extraction buffer (50 mmol/L sodium acetate, 1 mol/L NaCl, 0.1% Triton X100 10%, and acetic acid to pH of 4.0) with 20% (*v*:*v*) glycerol and protease inhibitors (Sigma-Aldrich, St. Louis, MO, USA; 11836153001), according to manufacturer’s instructions (Biosensis Pty Ltd., Thebarton, South Australia). In brief, tissues were sonicated (5–7 sec, ×2 30 min apart) in acid-extraction buffer at 10:1 [*w* (mg): *v* (µL)] on ice. Half of mixture was saved for immunoblotting, and the remaining was centrifuged (30 min, (10,000× *g*, 4 °C), to produce clarified supernatant. Total protein concentration was determined using Bradford assay [70] *Sample diluent* was prepared mixing 3:1 (*v*:*v*) acid-extraction buffer and incubation/neutralization buffer (1 mol/L phosphate buffer= 0.1 mol/L KH_2_PO_4_ + 0.1 mol/L Na2HPO4, pH 7.6) Clarified supernatants were diluted 1:1 (*v*:*v*) with *sample diluent* prior to incubating in duplicate Nrg1 antibody pre-coated well. A standard curve was produced by sequential dilution of lyophilized antigen in *sample diluent* (6.2 to 500 pg/mL). The detection antibody and streptavidin-HRP conjugate were diluted 100-fold with Assay Diluent A (from manufacturer). The 450 nm absorbance was read by a colorimetric plate reader with no incubation. Intra-assay and intraassay variability were 3.4% and 4.5% in our hands. 

### 4.7. Western Blotting (WB)

Methods were previously described [6,7,8]. In brief, homogenized protein (30 μg) was diluted in 4× loading buffer (3:1) and loaded into a 4–20% mini-protean TGX polyacrylamide precast gels (Bio-Rad Laboratories, Inc., Hercules, CA, USA). Protein transfer was achieved using TransBlot Turbo mid-size (Bio-Rad Laboratories, Inc.), and reliability was assessed using Ponceau S staining. Membranes were blocked with 2.5% bovine serum albumin (BSA) or NGS. Nitrocellulose blots were incubated (4 °C, overnight) in primary antibodies. After incubation, membranes were washed with TBS-T and then exposed to secondary antibodies at 1:10,000 in TBS-T for 1 h. Using enhanced chemiluminescence (Clarity Western ECL Substrate, Bio-Rad Laboratories, Inc), images were capture for optical density (OD) measured using ImageJ software (NIH, Bethesda, MD, USA) adjusted for background to quantify protein immunoreactivity relative to ponceau S [6,7].

#### Antibodies for WB (Appendix A)

(i)ErbB4 (Cell Signaling, 4795S; RRID: AB_2099883) Rabbit monoclonal IgG antibody raised against synthetic peptide corresponding to residues near the carboxy-terminus of human ErbB4 (1 μg/mL), detects a single band at ~150 kD(ii)phosphorylated ErbB4 (pErbB4) (ThermoFischer, PA5-12606; RRID: AB_10990353) Rabbit polyclonal IgG antibody raised against a KLH conjugated synthetic phosphopeptide corresponding to amino acid residues surrounding Tyr1162 of human ErbB4 (2 μg/mL), detecting a single band at ~150 kDa(iii)Akt (Cell Signaling, 2920S; RRID: AB_1147620) Mouse monoclonal IgG1 antibody raised against synthetic peptide at the carboxy-terminal sequence of human Akt (1 μg/mL), detecting a band at ~ 60kDa(iv)phosphorylated Akt (pAkt) (Cell Signaling, 9271S; RRID: AB_329825) Rabbit polyclonal IgG antibody raised against a synthetic phosphopeptide corresponding to residues surrounding Ser473 of mouse Akt (1 μg/mL), detecting a band at ~60 kDa(v)α-fodrin (Enzo Life Sciences, Farmingdale, NY, USA, FG6090; RRID: AB_2050678) Mouse monoclonal IgG1 antibody raised against chicken blood cell membranes following hypotonic lysis and mechanical enucleation (0.25 μg/mL), detecting bands at 250, ~150, and ~120 kDa(vi)GFAP (ProteinTech, 16825-1-AP; RRID: AB_2109646) Rabbit polyclonal IgG antibody raised against GFAP fusion protein (ProteinTECH, Ag10423) (0.2 μg/mL), detecting a band at ~50 kDa.

### 4.8. EcFP of Schaffer Collateral Synapse

Acute slices (300 μm) were prepared as detailed previously [71] in ice-cold dissection buffer (212.7 mM sucrose, 2.6 mM KCl, 1.23 mM NaH_2_PO_4_, 26 mM NaHCO_3_, 10 mM dextrose, 3 mM MgCl_2_, and 1 mM CaCl_2_ bubbled with 5% CO_2_/95% O_2_). The slices were recovered for 30 min at RT in ACSF (124 mM NaCl, 5 mM KCl, 1.25 mM NaH_2_PO_4_, 26 mM NaHCO_3_, 10 mM dextrose, 1.5 mM MgCl_2_, and 2.5 mM CaCl_2_ bubbled with 5% CO_2_/95% O_2_). Synaptic responses were evoked with 0.2 ms pulses (10–80 μA) delivered with ACSF-filled theta glass micropipettes, recorded extracellularly in CA1 from CA3 radiatum layer (Rd), and quantified as the initial slope of the field potential (FP). Baseline responses (0.033 Hz) used stimulation intensity that evoked a half-maximal response (maximal response without a population spike (pop-spike). Slices were discarded if pop-spike appeared in the initial rising phase (baseline hyperexcitability) or when the baseline was not stable (>5% drift). LTD was induced by 1 Hz stimulation every 30 sec over a period of 15 min, while LTP was induced using a single train of high frequency 100 Hz stimulation for 1 s. For each animal at P18–23 and P36–42, stimulus-evoked LTD and LTP were measured on the hippocampi ipsilateral (HI-exposed), contralateral (hypoxia alone), and sham controls.

### 4.9. Behavioral Testing

Testing was previously described in a quite experimental room between 6 am–8 am at low-light conditions [10,11]. The order of testing within each litter was randomized. 

#### 4.9.1. Open Field (P36)

Open field (OF) is used to assay locomotion, anxiety, and interest in new environments [72,73]. The mouse is placed into the center of the OF arena and can explore for 5 min. The time spent at the center was determined.

#### 4.9.2. Object Location Task (P36–P37)

Object location memory task (OLT) assesses spatial and working memory based on the premise that rodents spend more time exploring novel objects [74,75]. Twenty-four hours after habituation, the mouse is given 5 min to explore 4 novel objects, which will be explored again after 2 objects are swapped following a 30 min rest period. If the mouse was within 1 cm of the object and if its nose was pointed toward the object, behavior is deemed exploratory. The total time spent exploring objects (TT), the percentage of time exploring objects in familiar location (FE) and those in novel location (NE) were determined. An exploratory preference (EP) score was calculated (EP score = NE × 100/TT). Decreased EP suggests impaired working memory [76,77]. 

#### 4.9.3. Y-Maze (Phase 1 P34–35 & Phase 2 P38–P39)

Y-maze test is used to assess spatial working memory [78]. The plexiglass Y-maze apparatus shaped as the letter “Y,” has 3 arms in 120° angles. Working memory is assessed by recording the sequence of arm entries. Impaired working memory is determined by same arm returns in phase 1. Phase 2 is performed 3 d after phase 1 to determine spatial and recognition memory. The mouse explores the maze with one arm blocked (novel arm) for 5 min. After a 30 min rest period, the mouse explores all 3 arms of the maze and the percentage of time spent in the novel arm is determined.

### 4.10. Statistics

Since normality assumptions were not met, non-parametric Kruskal–Wallis test with post hoc Dunn-Bonferroni pair analysis for more than 2 groups or Mann–Whitney U test for 2 groups was applied at each separate time point. Non-electrophysiology results were presented as box and whisker plot. Electrophysiological traces are presented as a percentage change from baseline of FP slopes. Significance was assigned by *p*-value ≤ 0.05 in all cases. Non-parametric Spearman Rho correlation was used, and a best-fit regression line was calculated when appropriate. IBM SPSS Statistics 28v (IBM Corporation, Armonk, NY, USA) was used for analysis.

## 5. Conclusions

Neonatal HI leads to persistent deficits in hippocampal PV^+^ INs, which correlate with neurobehavioral deficits and disrupts mechanisms of synaptic plasticity. Mechanistically, acute decrease in hippocampal ErbB4 and Akt activation in response to neonatal HI may interrupt mechanisms of survival and maturation of PV^+^ INs. Late reemergence of ErbB4 deficits in the hippocampus may further impair PV^+^ INs maintenance long after the HI insult, and may play a role in the increased neurodegeneration that follows neonatal HI [6,7]. Additional studies are needed to better understand the bimodal changes in the ErbB4 activation in the hippocampus after neonatal HI and their direct consequences in GABAergic deficits and memory impairments to improve our ability to design delayed therapies to improve the outcomes of neonatal HI and TH.

## Figures and Tables

**Figure 1 ijms-24-00508-f001:**
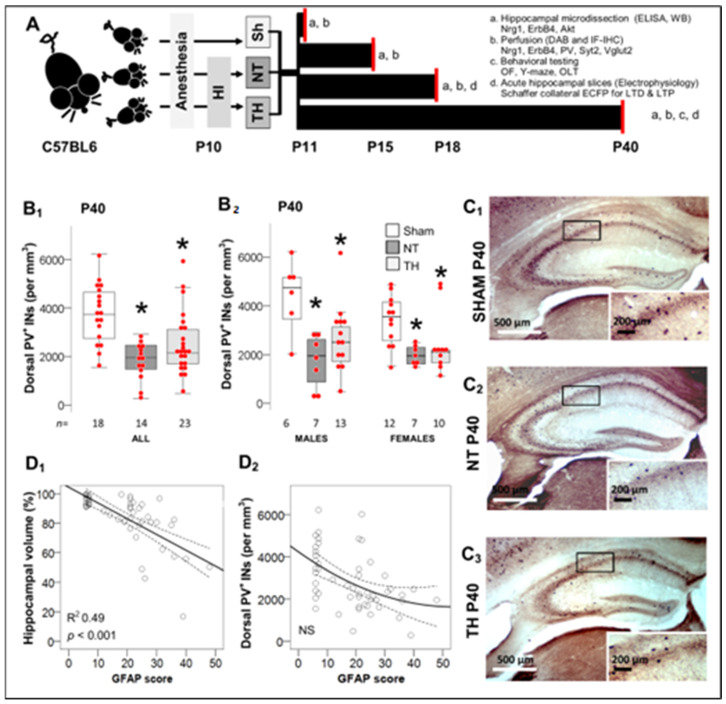
Experimental Design and Deficit of PV^+^ interneurons (INs) 30 days after HI. (**A**) P10 mice were randomized to sham (Sh) or hypoxic-ischemic (HI) injury with normothermia (NT) or therapeutic hypothermia (TH). Mice were aged to P11, P15, P18 or P40 for experimentation as detailed (a, b, c, and d). Neuropathology was only performed if hippocampus was not obliterated. Mice aged to P40 were used for behavioral experiments followed by neuropathology and biochemistry. Electrophysiology experiments (extracellular field potentials, EcFP), were performed in a separate group of mice at P18 and P40. (**B**) At P40, dorsal hippocampus from HI-injured mice had lower number of PV^+^ INs (per mm^3^) irrespective of treatment with NT or TH (**B_1_**) or sex (**B_2_**). Data are represented as hybrid box and whisker and dot plots, where box is limited by interquartile range (IQR), median is the line inside the box, and whiskers extend to the last datapoint within 1.5 × IQR from median. Outliers are outside the whisker boundaries. *, *p* < 0.05 (Dunn-Bonferroni post-hoc test for Kruskal Wallis- ANOVA). (**C**) Representative PV DAB-IHC of dorsal hippocampus at P40, (**C_1_**) SHAM P40, (**C_2_**) NT P40, (**C_3_**) TH P40. (**D**) GFAP score of severity of injury [6,7,8,35] linearly correlated with residual hippocampal volumes [8,9,11], (**D_1_**), but not with the number of PV^+^ INs (**D_2_**) using Spearman Rho correlation.

**Figure 2 ijms-24-00508-f002:**
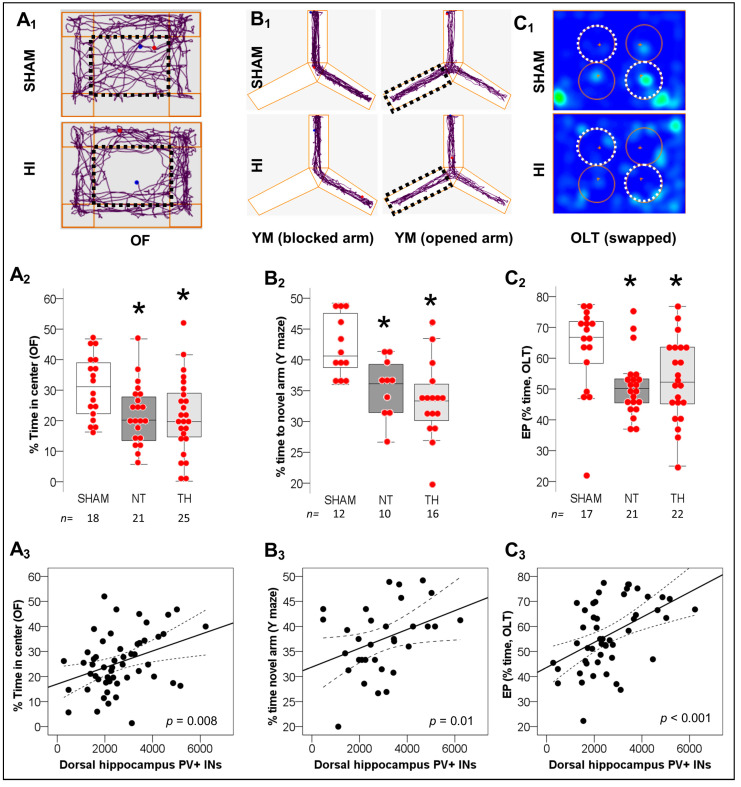
Behavioral Deficits after neonatal HI and correlation with deficits in hippocampal PV^+^ INs. (**A**) HI-injured mice spent less time in the center of the open field (OF) arena (discontinued line rectangle) compared to sham mice. (**B**) Similarly, HI-injured mice explores the novel arm (previously blocked, discontinued line rectangle) for longer % time after being opened in the Y-maze phase 2 (YM). (**C**) Following the swap of objects (discontinued line circles) during the object location task (OLT) phase 2, the exploratory preference (EP) for non-familiar location is decreased in HI-injured mice. Representative Anymaze Software tracings for OF (**A_1_**) and YM (**B_1_**) along with heatmap for OLT (more time = brighter/ red, less time = darker/ blue) (**C_1_**) are shown. Quantifications for OF (**A_2_**), YM (**B_2_**), and OLT (**C_2_**) are shown as hybrid box and whiskers with dot plots. *, *p* < 0.05 (Dunn-Bonferroni post-hoc test for Kruskal Wallis- ANOVA). The number of PV^+^ INs in the dorsal hippocampus directly correlated with decreased anxiety-like behavior in OF (**A_3_**) and improved working spatial memory in YM (**B_3_**) and OLT (**C_3_**), using Spearman Rho correlation test.

**Figure 3 ijms-24-00508-f003:**
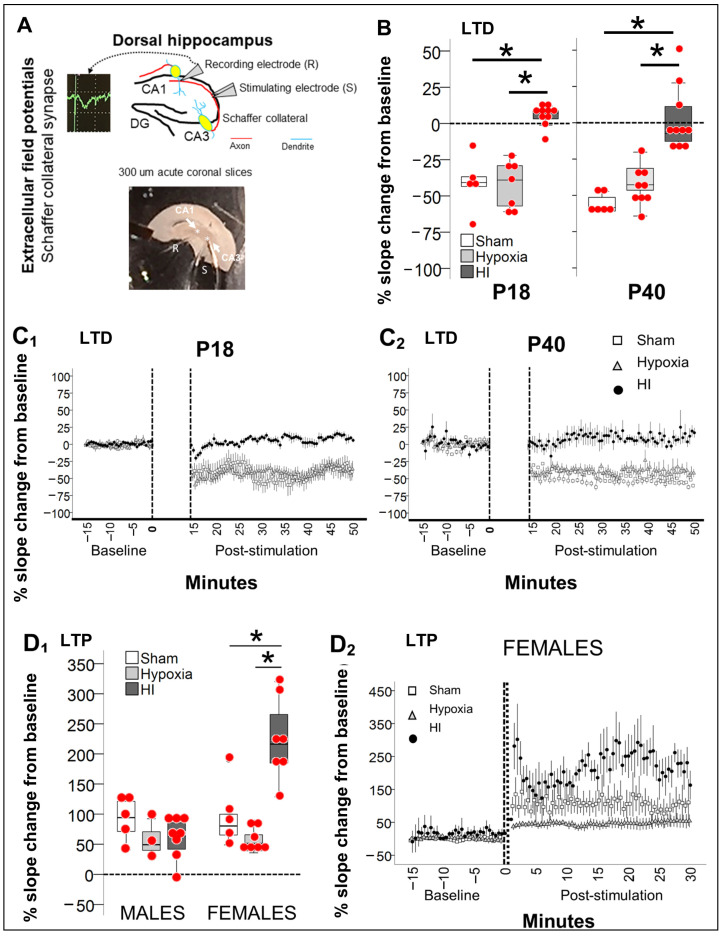
Disruption of the mechanisms of synaptic plasticity after neonatal HI. (**A**) EcFP of the Schaffer collateral synapse was performed in 300 µm-thick acute coronal slices containing dorsal hippocampus. Bipolar stimulating electrodes (S) were placed in the CA3 radiatum layer (Rd) and monopolar recording electrode (R) was placed in the CA1 Rd to detect extracellularly the percent change in slope of the field potential (FP), following stimulation to evoke long-term depression (LTD) and/or potentiation (LTP). Due to the developmental emergence of these mechanisms, EcFP was only performed after P18 for LTD and at P40 for LTP. (**B**,**C**) Evoked LTD was preserved in hippocampi from sham mice (white box in (**B**), open square in (**C**)) and hypoxia-alone exposed (contralateral to HI, light grey box in (**B**), and triangle in (**C**)), while LTD was blunted in HI-injured hippocampi (dark grey boxes in (**B**), and closed black circles in (**C**)) at both, P18 (**C_1_**) and P40 (**C_2_**). (**D**). In hippocampi from male mice, evoked LTP was equally preserved in sham (white box), and in hypoxia-alone (light grey) and HI-injured (dark grey, **D_1_**). In female mice, sham and hypoxia-alone hippocampi also demonstrated preserved evoked LTP, while HI-injured hippocampus had a heightened FP slope charge from baseline (**D_1_**,**D_2_**, closed circles). Quantifications are shown as hybrid box and whisker and dot plots. *, *p* < 0.05 (Dunn-Bonferroni post-hoc test for Kruskal Wallis ANOVA). Accumulative % change in FP slope from baseline (discontinuous lines represent time of stimulation) is shown per group over time for LTD (**C_1_**,**C_2_**) and for LTP (**D_2_**, only females).

**Figure 4 ijms-24-00508-f004:**
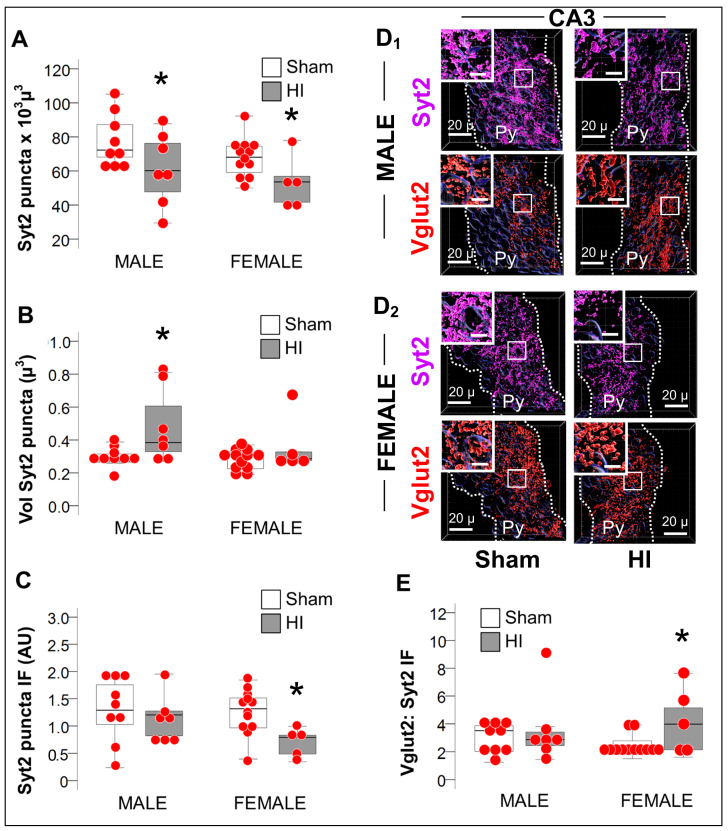
Decreased Syt2 puncta and predominance of excitation in female mice after HI. (**A**) The number of the puncta expressing the PV-specific presynaptic marker Syt2 was similarly decreased in the CA3 of both males and females HI-injured mice (grey box) compared to sham (white box). (**B**) Unlike, in female mice, CA3 Syt2^+^ puncta from HI-injured male mice (grey box) were larger in volume compared to sham (white box). (**C**) Syt2 immunofluorescence per puncta in arbitrary units (AU) was decreased in HI-injured female mice (grey box) compared to sham (white box), but not in male mice. (**D**) Representative Imaris reconstructions of Syt2 (magenta) and the glutamatergic Vglut2 (red) presynaptic markers within the pyramidal cell layer (Py) for male (**D_1_**) and female (**D_2_**) sham and HI-injured mice are shown. Radiatum (Rd) and Oriens (Or) layers were negated, to only account for perisomatic synapses within the Py. (**E**) The Vglut2:Syt2 ratio was greater in the HI-injured CA3 of female mice (grey box) supporting the EcFP LTP results shown in Figure 2D. Quantifications are shown as hybrid box, whisker and dot plots. *, *p* <0.05 (Mann-Whitney U test; *n*= 6, per sex per group).

**Figure 5 ijms-24-00508-f005:**
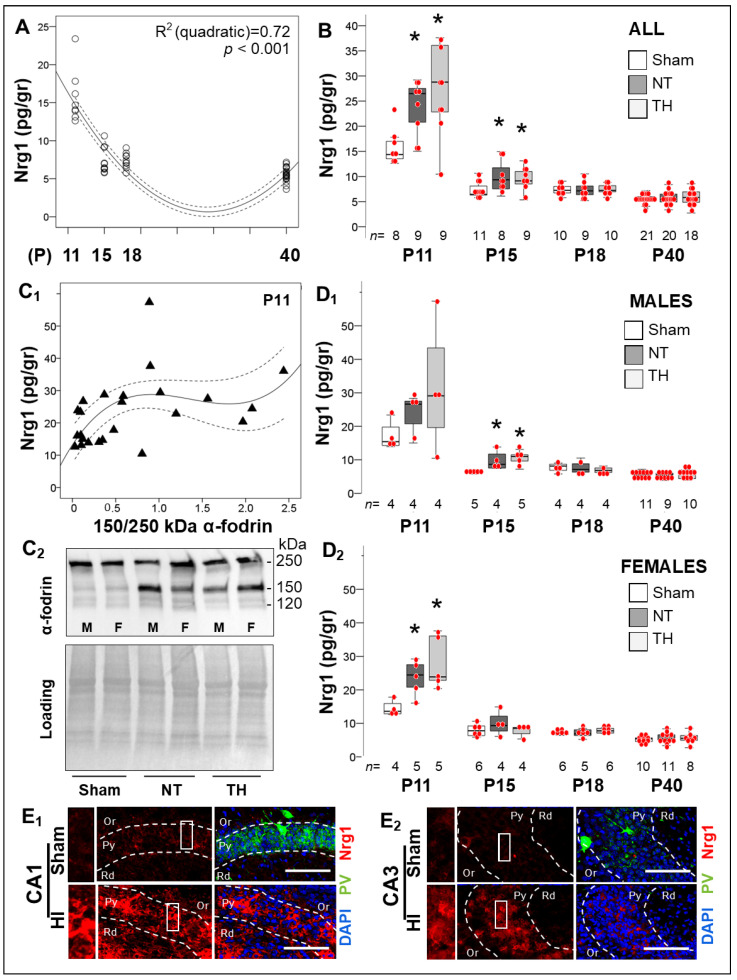
Long-term changes in Nrg1 levels in the hippocampus in response to neonatal HI. (**A**) Hippocampal Nrg1 levels decreased between P11 and P18 and reach a nadir prior to P40 (quadratic R^2^ 0.72, *p* < 0.001). (**B**) HI increased hippocampal Nrg1 levels by 84.4% (*p* = 0.01) and 44.8% (*p* = 0.05) at P11 and P15 after the insult, respectively (KW *p* < 0.05; *, Dunn-Bonferroni post-hoc *p* < 0.05 vs. sham). After 120h, Nrg1 was similar in all three groups. (**C**) The 150 kDa α- fodrin breakdown product weakly correlated with Nrg1 (Spearman’s Rho r = 0.53, *p* = 0.004) in a cubic regression, which plateaued at a Nrg1 level of ~28.5 pg/gr of protein (**C_1_**). Representative western blots demonstrating the breakdown of α- fodrin are shown at 120 and 150 kDa (**C_2_**). (**D**) Stratification by sex, demonstrated that Nrg1 peaked early after neonatal HI in both male and female mice (KW *p* = 0.02 and KW *p* = 0.01, respectively), but sooner in female mice. (**E**) Higher Nrg1 (red) and lower PV (green) expression are shown in CA1 (**E_1_**) and CA3 (**E_2_**) pyramidal cell layer (Py) in the dorsal hippocampus of HI-injured mice compared to sham by IF-IHC and confocal microscopy. Scale bar = 200 µm. Quantifications are shown as hybrid box & whisker and vertical dot plots. *, *p* < 0.05 (Dunn-Bonferroni post-hoc test for Kruskal Wallis ANOVA; *n* shown under each box).

**Figure 6 ijms-24-00508-f006:**
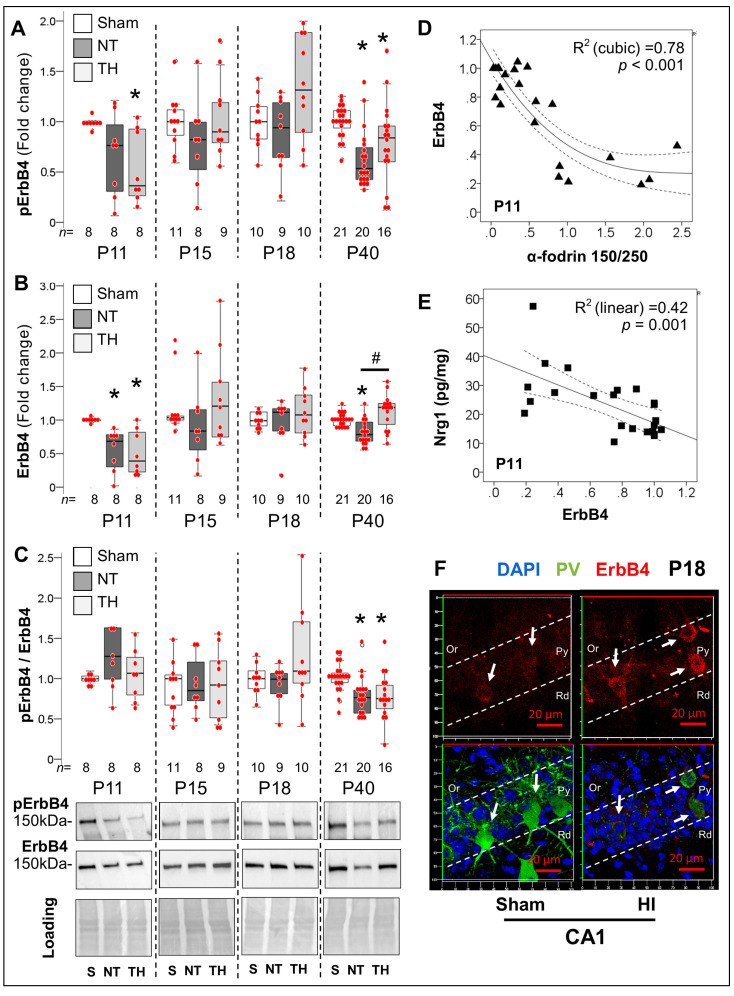
Acute and late deficits in ErbB4 receptor expression and activation after neonatal HI. (**A**,**B**) 24 h after HI ErbB4 phosphorylation (KW *p*= 0.04) and total expression (KW *p* = 0.001) decreased regardless of treatment with NT or TH. By P15 and P18, ErbB4 phosphorylation and total expression returned to levels similar to sham mice. Deficits remerged at P40 with a 43% (*p* <0.001) and 22% (*p* = 0.02) lower ErbB4 phosphorylation and total expression, respectively compared to sham (KW *p* < 0.001). TH prevented the decrease in total ErbB4 levels at P40 (*p* <0.001 vs. NT mice) but not the decrease in phosphorylation. (**C**) HI injury resulted in lower ErbB4 phosphorylated to total ratio (KW *p* = 0.001) at P40, regardless of treatment with TH. Representative blots are shown for Y1162 phosphorylated and total ErbB4 at 150 kDa, and ponceau S as loading control. (**D**) Hippocampal 150 kDa α-fodrin BDP relative to 250 kDa protein strongly predicted the decrease in ErbB4 expression at P11(R^2^
_(cubic)_ 0.78, *p* < 0.001). (**E**) The decrease in ErbB4 expression linearly predicted the increase in Nrg1 24 h after HI (R^2^ _(linear)_ 0.42, *p* = 0.001). (**F**) The recovery in ErbB4 expression after HI at P15 and P18 coincided with the increased expression (red) on surviving PV^+^ INs (green) in IF-IHC compared to sham. Quantifications are shown as hybrid box and whisker and vertical dot plots. *, *p* < 0.05 vs. sham; #, *p* < 0.05 vs. NT (Dunn-Bonferroni post-hoc test for Kruskal Wallis ANOVA; *n* shown under each box).

**Figure 7 ijms-24-00508-f007:**
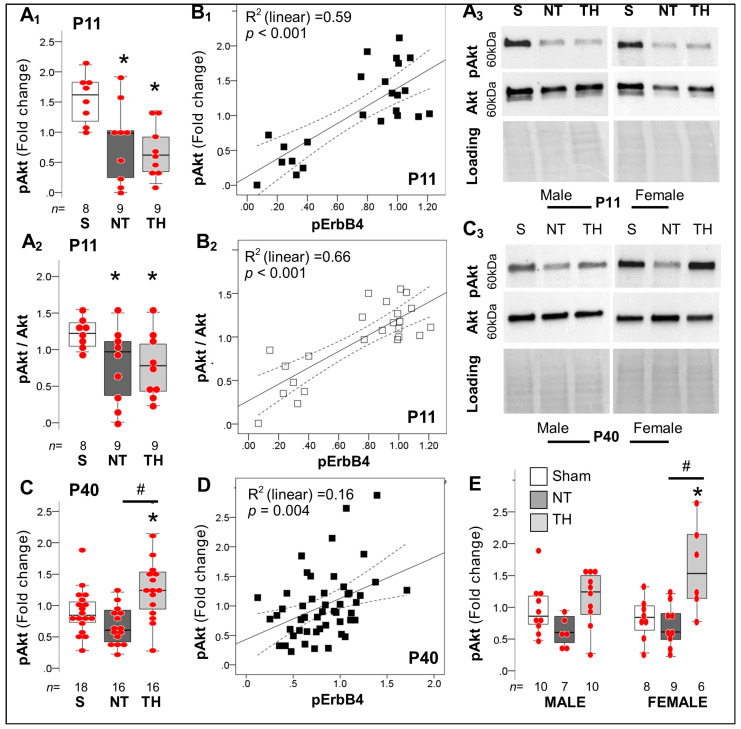
Alterations in downstream Akt phosphorylation in hippocampus after neonatal HI. (**A**) Phosphorylation of Akt was decreased in hippocampi from NT (*p* = 0.02) and in TH mice (*p* = 0.005) compared to sham at P11 (KW *p* < 0.01, **A_1_**), resulting in decrease ratio of Ser 473 phosphorylated to total Akt (**A_2_**). Representative blots show phosphorylated Ser 473 and total Akt at 60 kDa with Ponceau S as loading control. (**B**) Deficits in pAKT (**B_1_**) and the pAkt/ Akt ratio (**B_2_**) linearly correlated with the degree of Y1162 ErbB4 phosphorylation at P11. (**C**) At P40, HI injury did not produce significant deficits in AKT phosphorylation, but TH increased pAKT levels compared to sham and NT mice, respectively (KW *p* = 0.004). (**D**) ErbB4 phosphorylation weakly predicted the changes in pAkt at P40 (*p* = 0.004). (**E**) Sex stratification demonstrated increased pAkt most robustly in TH female mice. Quantifications are shown as hybrid box and whisker with vertical dot plots. *, *p* < 0.05 vs. sham; and ^#^, *p* < 0.05 vs. NT (Dunn-Bonferroni post-hoc test for Kruskal Wallis ANOVA; *n* shown under each box).

## Data Availability

Datasets will be made available upon direct request to corresponding author.

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
