# Peer review of "Dysregulation of ErbB4 Signaling Pathway in the Dorsal Hippocampus after Neonatal Hypoxia-Ischemia and Late Deficits in PV+ Interneurons, Synaptic Plasticity and Working Memory"

_ijms, 2022, doi:10.3390/ijms24010508_

Round 1

Reviewer 1 Report

1.The introduction is too brief. More detailed background information is needed.

2.The layout of the picture needs further optimization, and some of the acronyms and numbers should be explained in the picture notes. The labels of the pictures are unreasonable and inconsistent with the notes. Is it necessary to use A1 and A2?

Author Response

1.The introduction is too brief. More detailed background information is needed.

Response - Expanded upon points in the introduction while maintaining journal's recommendation to keep introduction brief.

2.The layout of the picture needs further optimization, and some of the acronyms and numbers should be explained in the picture notes. The labels of the pictures are unreasonable and inconsistent with the notes. Is it necessary to use A1 and A2?

Response: Picture layout has been optimized with further explanations included.

Reviewer 2 Report

It has been a pleasure to review this manuscript. Presented information appear clear, precise and well written. Extremely precision in laboratory results, excellent correlation with clinical outcomes. Congratulations to authors. Definitely, manuscript merits publication.

Author Response

Thank you

Reviewer 3 Report

The manuscript by Harisa Spahic et al., titled “Dysregulation of ErbB4 signaling pathway in the dorsal hippocampus after neonatal hypoxia-ischemia and late deficits in PV+ interneurons, synaptic plasticity and working memory” is an interesting study may be considered with major revision.

Minor comments:

1.      All the figure resolution need to be improved as they not readable in printed form

2.      In All the figure X and Y axis Font size should increase

3.      All the figure instead of box plot the data can be represented as dot plot

4.      All the figures, figure legend should simplify and more comprehensive

5.      In Figure 1. There are three groups and four bars are for P11-P40 is need to represent clear

6.      High resolution colored IHC image should be provided with a representative inset.

7.      Figure 3 please include data for P11, P15

8.      Figure 3 please include TH data

9.      Figure 4 please include TH and Hypoxia data

10.  Figure 4 and Figure 5 High resolution or larger size immunostaining images should be provided with a representative inset.

11.  Figure 5,6 and 7 Clear Western blot image with loading control can be provided and original uncropped images with protein marker should be provided in supplements with sufficient n- numbers.

12.  Please mention the molecular weight in western blot image Figure 6 and 7.  

Author Response

The manuscript by Harisa Spahic et al., titled “Dysregulation of ErbB4 signaling pathway in the dorsal hippocampus after neonatal hypoxia-ischemia and late deficits in PV+ interneurons, synaptic plasticity and working memory” is an interesting study may be considered with major revision.

  1. All the figure resolution need to be improved as they not readable in printed form

Response: Image quality has been optimized as requested.

  1. In All the figure X and Y axis Font size should increase

Response: it has been done.

  1. All the figure instead of box plot the data can be represented as dot plot

Response: A hybrid box and whisker with vertical dot plot has been used for all figures now.

  1. All the figures, figure legend should simplify and more comprehensive

Response: Pictures have been optimized and legends has been updated as requested by the reviewer.

  1. In Figure 1. There are three groups and four bars are for P11-P40 is need to represent clear

Response:  Panel A in Fig 1 has been modified as requested.

  1. High resolution colored IHC image should be provided with a representative inset.

Response:  Panels C in Fig 1 has been modified

  1. Figure 3 please include data for P11, P15

Response: Electrophysiology experiments at P11 and P15 were not performed due to the known developmental emergence of these mechanisms of synaptic plasticity in the hippocampal mouse. LTP and LTD cannot be readily elicited at P11 and P15 in part due to the incomplete myelination and speed of propagation from CA3 to CA1. A brief explanation has been added to the legend to Fig 3.

  1. Figure 3 please include TH data

Response: Since TH did not provide significant protection against deficits in PV+ INs count and behavioral performance, as showed in Fig 1 and 2, electrophysiology experiments were focused in the understanding of the effects of HI injury using sham hippocampus (separate set of animals) and hypoxia-alone exposed hippocampus (left hemisphere, contralateral to the HI injury) as controls for comparisons.  The goal of these experiments was to find an electrophysiological correlate to the behavioral deficits shown in figure 2, thus these complicated experiments were hypothesis-driven. We have added explanation in lines 105 to 109.

  1. Figure 4 please include TH and Hypoxia data

Response: Similar to Fig 3, these IHC experiments were performed to understand the effect of HI in the Vglut2: Syt2 at P40 to better complement the behavioral results shown in Fig 2. Once again TH, had no effect in attenuating the behavioral outcomes, thus, this group is not the focus of the experiments presented in Fig 4.

  1. Figure 4 and Figure 5 High resolution or larger size immunostaining images should be provided with a representative inset.

Response: Changes in figures 4 and 5 had been made as requested.

  1. Figure 5,6 and 7 Clear Western blot images with loading control can be provided and original uncropped images with protein marker should be provided in supplements with sufficient n- numbers.

Response: Uncropped blots had been added as supplemental material. 

  1. Please mention the molecular weight in western blot image Figure 6 and 7.  

Response: Molecular weight has been added for those missing.

Round 2

Reviewer 3 Report

Congratulation!

Please include all the WB data  from non-published to supplementary file